# Biologic Treatment in Combination with Lifestyle Intervention in Moderate to Severe Plaque Psoriasis and Concomitant Metabolic Syndrome: Rationale and Methodology of the METABOLyx Randomized Controlled Clinical Trial

**DOI:** 10.3390/nu13093015

**Published:** 2021-08-29

**Authors:** Andreas Pinter, Peter Schwarz, Sascha Gerdes, Jan C. Simon, Anja Saalbach, James Rush, Nima Melzer, Thomas Kramps, Benjamin Häberle, Maximilian Reinhardt

**Affiliations:** 1Department of Dermatology, Venereology and Allergology, Clinical Research Division University Hospital Frankfurt am Main, Theodor-Stern-Kai 7, 60590 Frankfurt am Main, Germany; 2Department for Prevention and Care of Diabetes, Medical Faculty Carl Gustav Carus, Technical University of Dresden, 01062 Dresden, Germany; peter.schwarz@uniklinikum-dresden.de; 3Paul Langerhans Institute Dresden of the Helmholtz Center Munich at University Hospital and Faculty of Medicine, 01307 Dresden, Germany; 4German Center for Diabetes Research, 85764 Neuherberg, Germany; 5Psoriasis-Center, Department of Dermatology, Campus Kiel, University Medical Center Schleswig-Holstein, 24105 Kiel, Germany; sgerdes@dermatology.uni-kiel.de; 6Department of Dermatology, Venerology and Allergology, Leipzig University Medical Center, 04103 Leipzig, Germany; Jan-Christoph.Simon@medizin.uni-leipzig.de (J.C.S.); Anja.Saalbach@medizin.uni-leipzig.de (A.S.); 7Novartis Pharma AG, 4056 Basel, Switzerland; james.rush@novartis.com (J.R.); maximilian.reinhardt@novartis.com (M.R.); 8Novartis Pharma GmbH, 90429 Nuremberg, Germany; nima.melzer@novartis.com (N.M.); thomas.kramps@novartis.com (T.K.); benjamin.haeberle@novartis.com (B.H.)

**Keywords:** psoriasis, obesity, metabolic syndrome, secukinumab, inflammation

## Abstract

Inflammatory diseases including psoriasis are associated with metabolic and cardiovascular comorbidities, including obesity and metabolic syndrome. Obesity is associated with greater psoriasis disease severity and reduced response to treatment. Therefore, targeting metabolic comorbidities could improve patients’ health status and psoriasis-specific outcomes. METABOLyx is a randomized controlled trial evaluating the combination of a lifestyle intervention program with secukinumab treatment in psoriasis. Here, the rationale, methodology and baseline patient characteristics of METABOLyx are presented. A total of 768 patients with concomitant moderate to severe plaque psoriasis and metabolic syndrome were randomized to secukinumab 300 mg, or secukinumab 300 mg plus a tailored lifestyle intervention program, over 24 weeks. A substudy of immunologic and metabolic biomarkers is ongoing. The primary endpoint of METABOLyx is PASI90 response at week 24. Other endpoints include patient-reported outcomes and safety. METABOLyx represents the first large scale clinical trial of an immunomodulatory biologic in combination with a standardized lifestyle intervention.

## 1. Introduction

Immunology and metabolism were traditionally regarded as two different physiological entities. However, they are evolutionarily very closely connected, as exemplified in simpler organisms such as *Drosophila melanogaster*, where the fat body carries out both metabolic and immune functions [1]. The role of the human immune system in primarily non-immune processes such as metabolism is now evident, with inflammation understood as a key driver of metabolic processes and diseases, giving rise to the new scientific field of immunometabolism [2]. Conversely, metabolic factors can significantly impact inflammatory processes and diseases. For example, adipose tissue and liver inflammation drive insulin resistance in type 2 diabetes, and endothelial inflammation drives plaque formation in coronary artery disease [3,4].

Psoriasis, as an immune-mediated inflammatory disease, is an ideal model for the study of this interconnection between inflammation and metabolism. As a consequence of systemic inflammation, psoriasis patients have an elevated risk of metabolic and inflammatory comorbidities including obesity, type 2 diabetes, dyslipidaemia and cardiovascular disease and metabolic syndrome [5,6,7]. Metabolic syndrome is defined as a concurrence of abdominal obesity, dyslipidaemia, hypertension or insulin resistance, which significantly increases the risk of cardiovascular disease, stroke and type 2 diabetes [8,9]. Insulin resistance, a criterion of the metabolic syndrome, is also observed as a component of psoriasis pathophysiology (the ‘psoriatic march’ concept) [10,11]. This association can also impact the clinical management of psoriasis; for example, higher body weight and body mass index (BMI) are associated with diminished response to biologic treatment and reduced length of remission [12].

Obesity is known to predispose patients to psoriasis, and to amplify psoriatic inflammation, and conversely, psoriatic inflammation exacerbates the metabolic comorbidities of obesity [13,14,15,16]. In obesity, the expanded adipose tissue, in particular the visceral adipose tissue, enters a state of low-grade chronic inflammation, termed meta-inflammation [17,18]. During meta-inflammation, adipose tissue becomes infiltrated by activated immune cell populations, causing the expression of inflammatory cytokines, including TNF-α or IL-17 [19,20]. The increased expression of pro-inflammatory cytokines in the inflamed adipose tissue microenvironment leads to higher local and systemic cytokine concentrations, which can be measured in the serum of obese individuals (Figure 1) [21,22,23]. Both psoriasis and obesity are therefore inflammatory diseases that contribute to systemic inflammation. 

Systemic inflammatory processes and metabolic status are also closely linked to bone health. TNFα and IL-17, central to the pathogenesis of psoriasis, are known to impair bone formation [24]. However, the HUNT3 study found no association between psoriasis and bone fracture risk, bone mineral density, T-score or higher prevalence of osteoporosis [25]. Low-grade inflammation mediated by these factors may also influence bone metabolism in obese patients [24]. Procollagen-1 N-terminal peptide (P1NP), a marker of bone and osteoporosis risk, may also be altered in patients with metabolic syndrome, and also in the inflammatory context of non-alcoholic fatty liver disease (NAFLD) [26,27]. However, little is known of the pathophysiological mechanisms and consequences of P1NP dysregulation in psoriasis. Links between metabolic syndrome and bone metabolism also remain to be fully explored [28].

Secukinumab is a fully human monoclonal antibody that selectively neutralizes IL-17A, a cytokine fundamental to the development of psoriasis [29]. Secukinumab has shown long lasting efficacy and safety in the treatment of moderate to severe plaque psoriasis [29,30,31]. Links have been established between obesity and metabolic comorbidities and the outcomes of psoriasis treatment with secukinumab and other biologics [12]. 

Obesity and other metabolic comorbidities may be factors characterizing patients with inadequate responses to psoriasis treatment. A post hoc analysis of pooled secukinumab phase III trial data revealed that response rates were higher as mean body weight, waist circumference, and BMI decreased [32]. A separate post hoc analysis showed that psoriatic arthritis, metabolic syndrome, obesity, impaired glucose metabolism/insulin resistance, and hyperuricemia were each associated with increased hs-CRP levels at baseline [30]. Metabolic syndrome showed an impact on systemic inflammation similar to psoriatic arthritis [33]. Secukinumab treatment reduced the levels of hs-CRP; however, concomitant obesity attenuated this decline in hs-CRP under treatment [33]. In a pooled analysis, increased levels of the anti-inflammatory adipose tissue hormone adiponectin were associated with metabolic syndrome and obesity; however, psoriasis severity and anti-psoriatic treatment with secukinumab or etanercept had no relevant impact on adiponectin levels [34]. In addition, systemic inflammation and pro-inflammatory IL-17 signalling were shown to persist at the end of treatment in psoriasis patients with metabolic syndrome, reducing the length of remission and resulting in earlier relapse compared to psoriasis patients without metabolic syndrome [35]. 

Hyperuricemia is also increasingly understood as a key cardiovascular risk factor and mediator of endothelial inflammation, and is associated with both psoriasis and obesity [36]. Secukinumab is shown to reduce systemic inflammation as measured by hsCRP [33] and was also found to potentially reduce cardiovascular risk, as assessed by flow-medicated dilation, a measure of endothelial function [37]. However, a study of treatment with secukinumab showed that hypertriglyceridemia was still present in patients despite improved psoriatic disease activity after secukinumab treatment [38]. 

These previous analyses suggest that treating skin inflammation alone is not sufficient in psoriasis patients with concomitant metabolic syndrome. A treatment approach focused solely on the skin does not address obesity and metabolic factors as significant sources of systemic inflammation, with clinical consequences including high levels of systemic inflammation, lower treatment efficacy for the skin, higher chance of relapse, and higher cardiometabolic risk [33,34,38]. The evidence indicates an opportunity for a holistic treatment strategy for psoriasis patients with metabolic comorbidities, one that addresses skin and joint manifestations, but that also targets metabolic diseases and the inflammation associated with them. Modifiable lifestyle factors, in particular weight reduction strategies, are known to positively impact psoriasis symptoms including reduction in PASI [39]. To date, small studies have shown improvements in response to systemic and biologic treatments with weight loss [40,41]. A meta-analysis of 10 randomized controlled trials of lifestyle interventions involving diet and/or exercise and health education in patients with psoriasis concluded that such interventions may reduce the severity of psoriasis, but the quality of evidence was low [42].

The METABOLyx trial (EudraCT no. 2016-001671-79, NCT03440736) was designed to evaluate the impact of combining a lifestyle intervention program with secukinumab treatment in patients with moderate to severe plaque psoriasis and metabolic syndrome in Germany. METABOLyx is the first large scale randomized controlled trial to compare lifestyle interventions as an addition to biologic therapy and to examine effects on cardiometabolic outcomes. Given the known association of psoriasis severity with obesity, the hypothesis of METABOLyx is that the combination of secukinumab with lifestyle intervention would improve both skin symptoms and cardiometabolic status, more than secukinumab alone, by dual targeting of the shared pathophysiology behind both diseases, i.e., systemic inflammation. An extensive biomarker substudy will also form an integral part of the trial, dissecting the mechanisms of inflammation and immune activation and characterizing the metabolic profiles of patients. Serum levels of markers of bone formation will also be measured, and will be interpreted in the context of a standard liver function panel and other biomarkers. The biomarker data will then be correlated with clinical outcome measures, metabolic disease markers, including free fatty acids and triglycerides, and the liver function panel. 

## 2. Materials and Methods

### 2.1. Study Design and Patients

The METABOLyx trial is a randomized, controlled, multicentre study with two open-label treatment arms. METABOLyx recruited adult (aged ≥ 18 years) patients with moderate to severe plaque psoriasis (PASI > 10) for 6 months or more and concomitant metabolic syndrome at screening as previously defined by international consensus (Table 1) [8]. Patients who did not qualify for dietary changes and increased physical activity because of any medical condition, such as significant cardiac comorbidities or any other medical conditions that would put them at risk when participating in the lifestyle intervention, were excluded from the study. A full list of eligibility criteria and concomitant medication use is given in Appendix A. Written informed consent was obtained from all subjects. The METABOLyx study is being conducted in line with the principles of the Declaration of Helsinki and the International Conference on Harmonization Good Clinical Practice guidelines. The study protocol was approved by the ethical review board of participating centres.

Patients were randomized 1:1 to one of two treatment arms in the core study. Patients in arm A will receive secukinumab 300 mg (approved dose; weekly from weeks 0 to 4 then at 4-weekly interval) from week 0 until week 24. Patients in arm B will receive secukinumab 300 mg from week 0 until week 24 and, in addition, participate in a concomitant lifestyle intervention program (Figure 2). Following week 28, there will be an extension period to week 56 where all patients may participate in the lifestyle intervention program (Figure 2), in conjunction with any psoriasis therapy as deemed appropriate by the treating physician. The purpose of this is to collect long-term data up to one year and offer the lifestyle intervention to all patients.

The structured and standardized lifestyle intervention program was designed specifically for the METABOLyx study by an interdisciplinary team, including a nutritionist, psychologist, sports scientist and specialists in internal medicine, diabetology and dermatology, (TUMAINI Institute Dresden, Germany; Department of Internal Medicine, Technical University Dresden, Germany). The program was developed based on the effectiveness of lifestyle interventions leading to weight loss in the prevention and treatment of type 2 diabetes and its complications [43]. The program goal is to support psoriasis patients with concomitant metabolic syndrome to improve their metabolic status, increase their physical activity level, change their diet and to lose weight. Patients in arm B (or after 24 weeks of secukinumab treatment in arm A) will be individually and regularly supervised over eight scheduled 60-min visits by nominated trainers associated with participating study centres. All trainers receive standardized instruction and work from a manual, covering patient motivation, action planning, interventions including dietary changes, stress management, and relapse prevention. Trainers receive their own on-site or telephone supervision sessions every four weeks. All interventions are evidence-based and tailored to patients’ needs. Patients will receive written standardized educational material on nutrition, physical activity and metabolism, including advice on changing their diet and increasing their physical activity level. Connected digital devices to measure physical activity (Activity SensorAS 80, Beurer or Walking Style VI, Omron) as well as body weight (scale, GS 485, Beurer) are given to patients. The measurement data can be continuously tracked by patients through an app (HealthManager app, Beurer). Patients are also instructed to use study workbooks. Key targets of the lifestyle intervention are given in Table 2. Patient weight, blood pressure, waist circumference and PASI score will be taken at each visit to monitor progress; in addition, lipid profile, hsCRP and HbA1c will be measured at the first and final visits.

The exploratory biomarker substudy will allow comprehensive characterization of the patient´s immunologic and metabolic profile from assessment of the markers detailed in Table 3. Samples for the biomarker substudy will be taken at baseline, week 16 and week 28 from 50 patients in arm A and 50 patients in arm B. 

### 2.2. Endpoints

The primary endpoint of the study is the percentage of patients achieving PASI90 at week 28 in each arm (by blinded assessor rating of skin improvement). The hypothesis of METABOLyx is that secukinumab combined with a lifestyle intervention results in a higher PASI 90 response rate in psoriasis patients with metabolic syndrome than secukinumab alone. Secondary endpoints include assessments of absolute reduction in PASI between arms, measures of systemic inflammation including hs-CRP; cardiometabolic markers including fasting plasma glucose, HbA1c, HDL, LDL and blood pressure; and weight, BMI and waist circumference. Steps per day as a measure of physical activity will also be recorded as a secondary endpoint. Dermatology life quality index (DLQI), WHO-5 and assessments of itch, pain and scaling will also be recorded. The exploratory endpoints of the biomarker substudy include levels of selected biomarkers (Table 3) throughout treatment and any association with clinical response. Safety assessments will be carried out at each study visit. Adverse events will be reported by investigators and by lifestyle intervention trainers. The adverse event profile of secukinumab is well established for the approved dose [30].

### 2.3. Statistical Analysis

METABOLyx was designed to demonstrate superiority of secukinumab combined with a lifestyle intervention vs. secukinumab alone in terms of PASI90 response at week 28. A PASI90 response of 81% at week 28 under secukinumab alone in patients with metabolic syndrome is assumed based on analysis of previous Phase III trial data [32]. Evidence on the effect size of lifestyle intervention on PASI response is relatively sparse. Assumptions are based on clinical studies showing an absolute increase of 6% to 27% in PASI75 response to biologic treatment in overweight or obese, moderate to severe psoriasis patients undergoing weight reduction compared to patients not undergoing weight reduction [39,40]. Assuming an absolute increase of 9% in PASI90 response in arm A compared to arm B, 342 patients per arm would provide a power of 90% at a two-sided alpha of 0.05 to demonstrate that the percentage of PASI90 responders at week 28 is higher in the secukinumab combined with lifestyle intervention arm compared to the secukinumab alone arm. Superiority will be tested on an intention to treat (ITT) basis using a logistic regression model with the factors treatment, centre and covariate baseline PASI. The odds ratio and its 95% confidence interval (CI) and *p*-value will be given. To compensate for expected drop-outs and/or premature discontinuations, recruitment of 380 patients per arm is planned for this trial. Patients who do not have a valid PASI assessment at week 28 will be regarded as non-responders for the primary analysis. Non-responder imputation will also be applied to all secondary response variables.

## 3. Results

### Patient Baseline Characteristics

A total of 768 patients were enrolled in METABOLyx and randomised to secukinumab 300 mg (Arm A, *n* = 366) or secukinumab 300 mg plus lifestyle intervention (Arm B, *n* = 402; difference in numbers due to randomization block size). Baseline characteristics were balanced between groups, with both groups having a high baseline PASI (19.7) and reporting significant impact on quality of life by DLQI (Table 4).

## 4. Discussion

The METABOLyx trial was designed to assess whether a combination of secukinumab and a lifestyle intervention improves skin symptoms and cardiometabolic status more than secukinumab alone in patients with psoriasis and concomitant metabolic syndrome. METABOLyx aims to provide a proof-of-concept for the integrated treatment of both psoriasis and metabolic disease. Previous small-scale studies have shown a positive impact of weight loss and lifestyle interventions on psoriasis treatment outcomes and cardiometabolic parameters [39,40,41]. To the authors’ knowledge, this is the first time an immunomodulatory biologic is combined with a lifestyle intervention in a large scale, randomized and controlled clinical trial of novel design.

Secukinumab is effective in treating psoriatic skin lesions and psoriatic arthritis [30,43]. However, to optimally treat patients with psoriasis and concomitant metabolic syndrome, given the close interconnection of both conditions, an integrated treatment approach addressing all sources of inflammation is needed [12,21,32]. Given the overlapping inflammatory pathophysiology behind psoriasis and cardiometabolic disease, it is hypothesized that secukinumab and lifestyle intervention would act synergistically, improving both conditions to a greater extent than each treatment alone. PASI90 was therefore chosen as the primary clinical endpoint to assess the effectiveness of lifestyle interventions on psoriasis severity, with metabolic markers as secondary endpoints. Post hoc analyses have also revealed that while secukinumab treatment lowers levels of systemic inflammation as measured by hs-CRP, this effect is diminished in the presence of obesity and metabolic syndrome [33]. The impact of treatment with secukinumab and other biologics on other cardiometabolic risk factors, such as hypertension, adiponectin levels, dyslipidaemia and hyperuricemia, is also limited in patients with metabolic comorbidities [33,35]. This results in a continued presence of cardiovascular and metabolic risk in these patients despite psoriasis treatment. 

Evidence based lifestyle interventions were selected for application in METABOLyx, including dietary intervention, increased physical activity, and face-to-face supervision and coaching. METABOLyx aims for at least 5% body weight loss over 6 months; this was considered achievable if patients are adherent to the intervention program. A previous study of a lifestyle intervention in type 2 diabetes showed that 50% of patients achieved weight reduction of ≥7% at 6 months, with a self-reported adherence to the program of 74% [43].

While the interplay of proinflammatory factors leading to co-association of psoriasis and obesity is well-established, other factors are likely to contribute additionally to the high rates of obesity observed in psoriasis patients. These include depression, hypercaloric nutrition, alcohol abuse, social isolation and lack of exercise, often associated with lower socioeconomic status [44]. By recording DLQI and other patient-reported outcomes, METABOLyx will provide insight into how some of these quality-of-life factors might be influenced by effective psoriasis treatment with or without lifestyle intervention.

The planned biomarker substudy aims to assess the metabolic and immune profiles of the patients and correlate them with the clinical status and treatment outcomes. The availability of a very large clinical dataset including data on psoriasis severity, treatment efficacy, and comorbidities in 760 patients, in combination with comprehensive molecular phenotyping of their metabolic and immune status, will enable a deeper understanding and the dissection of the interconnection between psoriasis and metabolic disease. 

Potential limitations of the METABOLyx study include the necessity of the lifestyle intervention arm being unblinded and open-label. Adherence to lifestyle interventions is challenging and high drop-out rates have been observed in previous studies of lifestyle interventions [42]. To increase adherence to the lifestyle intervention the design of the METABOLyx lifestyle intervention program incorporates regular face-to-face visits with a trainer, to encourage adherence and support patient motivation. In addition, connected digital devices to measure physical activity and body weight, and track data continuously via an app, are part of the study.

Viewing metabolism and the immune system as closely interconnected entities has been applied in various medical specialties in recent years and has contributed to advances in understanding and treatment of various diseases. A central aim of the METABOLyx trial is to apply this idea to psoriasis and translate it into a more integrated and effective treatment strategy. Furthermore, the study aims to contribute to a deeper understanding of the immune-metabolic intersections that are present in this disease. The combination of secukinumab and a lifestyle intervention program is currently unique, and could have significant implications for clinical management of people with chronic inflammation, psoriasis and/or metabolic syndrome. Ultimately, this could result in an improved treatment strategy for psoriasis patients, taking into account the full complexity of their disease, and could potentially decrease the risk both of the development of aspects of metabolic syndrome and of metabolic complications. In the context of the SARS-CoV-2 (COVID-19) pandemic, where obesity and diabetes are emerging as risk factors for adverse outcomes [45], these interventions become even more relevant.

## 5. Conclusions

In summary, METABOLyx represents the first assessment of an immunomodulatory biologic in combination with a standardized lifestyle intervention in a large scale, randomized, controlled clinical trial. This integrated approach targets inflammatory processes from multiple angles in order to achieve optimal, holistic treatment in patients who exhibit a complex, interconnected group of diseases.

## Figures and Tables

**Figure 1 nutrients-13-03015-f001:**
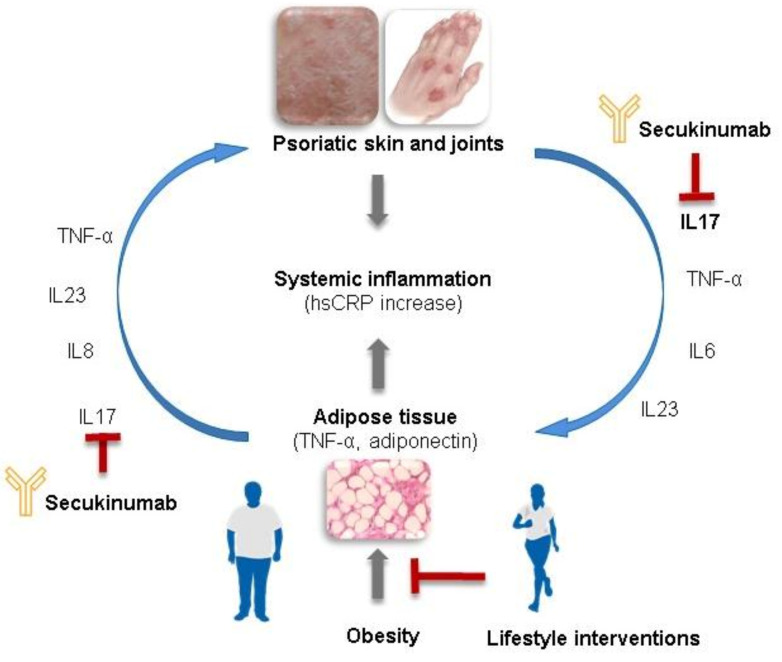
Pathophysiology of psoriasis and obesity. The pathophysiological processes of systemic inflammation that underlie psoriasis and obesity/metabolic syndrome are interconnected via the release of inflammatory mediators from both psoriatic lesions and adipose tissue. IL-17 blockade with secukinumab inhibits aspects of this inflammatory cycle, but lifestyle interventions aimed at reducing obesity and associated inflammation may produce synergistic effects.hsCrp, high-sensitivity C-reactive protein; IL, interleukin; TNF, tumour necrosis factor.

**Figure 2 nutrients-13-03015-f002:**
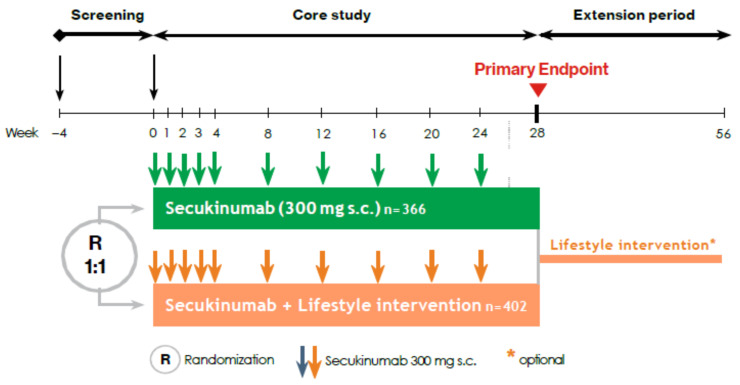
METABOLyx study design. METABOLyx is an open-label randomized controlled trial comparing secukinumab 300 mg (arm A) to secukinumab 300 mg plus a standardized lifestyle intervention (arm B). Primary endpoint is PASI90 response at week 28; secondary endpoints include metabolic syndrome components and safety. An exploratory biomarker substudy will also be conducted to obtain a profile of patients’ metabolic and inflammatory/immune status throughout treatment. During the extension period, lifestyle intervention can be continued by patients who have been in arm B during the core study or can be started by patients who have been in arm A. Participation in the lifestyle intervention during the extension period is not mandatory. Participation in the extension period itself is mandatory. Psoriasis treatment during the extension period can be chosen freely by the investigator.

**Table 1 nutrients-13-03015-t001:** Definition of metabolic syndrome [8].

Patients Meeting 3 Out of the 5 Criteria Meet the Diagnostic Requirements for Metabolic Syndrome
Fasting (8 h) plasma glucose ≥100 mg/dL or ongoing antidiabetic drug treatment. Abdominal obesity defined by elevated waist circumference: Male: ≥94 cm, female: ≥80 cm (except for patients of Asian, South or Central American ethnicity, for whom the cut off values are: Male: ≥90 cm, female: ≥80 cm). Fasting (8 h) triglycerides ≥ 150 mg/dL or ongoing drug treatment for elevated triglycerides. Fasting (8 h) HDL-C < 40 mg/dL in men or <50 mg/dL in women or ongoing drug treatment for reduced HDL-C. Resting blood pressure: Systolic blood pressure ≥ 130 and/ or diastolic blood pressure ≥ 85 mmHg or ongoing antihypertensive drug treatment.

**Table 2 nutrients-13-03015-t002:** Lifestyle intervention targets.

Key Components of METABOLyx Lifestyle Intervention Program
▪ Weight reduction ≥ 5%. ▪ Increase in physical activity (to a total of at least 30–60 min of activity day). ▪ Increasing the proportion of fibre in food intake ≥ 15 g. ▪ Reduction in the fat content of daily food intake < 30%. ▪ Reduction of saturated fatty acid intake < 10%.

**Table 3 nutrients-13-03015-t003:** Biomarkers assessed in METABOLyx substudy.

Classification	Markers Analysed
Inflammation	IL-6, TNF-alpha, IL-1 beta, IL-1Ra, IL-18, CD154
Metabolic profile	Chromatography based full free fatty acid serum profiles, fasting insulin, HOMA-IR, proinsulin, adipokine profile (adiponectin, leptin)
Liver status	M30 assay
Bone metabolism	P1NP, CTX, RANKL, OPG, sclerostin, sThy-1

IL-6, interleukin-6; IL-1 β, interleukin-1 β; IL-18, interleukin-18; CD154, cluster of differentiation 154; HOMA-IR Homeostasis Model Assessment of Insulin Resistance; M30 assay, enzyme-linked immunosorbent assay developed for the detection of soluble caspase-cleaved keratin 18; P1NP, Procollagen 1 N Terminal Propeptide; CTX, C-terminal telopeptide; RANKL, Receptor activator of nuclear factor kappa-Β ligand; sThy-1, soluble Thy-1 cell surface antigen.

**Table 4 nutrients-13-03015-t004:** Baseline characteristics of METABOLyx study population (*n* = 768).

Baseline Characteristics	Secukinumab 300 mg (A) (*n* = 366)	Secukinumab 300 mg + Lifestyle Intervention (B) (*n* = 402)
Age, years, mean (SD)	50.5 (13.2)	50.1 (12.5)
Gender (Male) *n* (%)	263 (71.7)	289 (71.7)
Race *n* (%)		
Caucasian	355 (96.7)	391 (97.0)
Other	2 (0.5)	8 (2.0)
Asian	8 (2.2)	1 (0.2)
Black	2 (0.5)	2 (0.5)
Unknown	0 (0.0)	1 (0.2)
Weight kg, mean (SD)	107.13 (22.57)	106.74 (20.58)
BMI kg/m^2^, mean (SD)	34.82 (6.83)	34.65 (6.45)
Waist circumference cm, mean (SD)	115.54 (15.41)	114.99 (14.06)
Cardiovascular history, *n* (%)		
Hypertension	241 (44.9)	272 (47.3)
Dyslipidaemia/Hyperlipidaemia	146 (27.2)	150 (26.1)
Type 2 diabetes	86 (16.0)	78 (13.6)
Baseline PASI, mean (SD)	19.76 (8.01)	19.72 (7.29)
Baseline DLQI, (0–30) mean (SD)	17.18 (6.39)	16.79 (6.84)
Prior psoriasis treatment, *n* (%)		
Topical	304 (44.6)	330 (46.5)
Non-biologic systemic therapy	176 (25.8)	185 (26.1)
Biologic systemic therapy	26 (3.8)	25 (3.5)

## Data Availability

Not applicable.

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
