# Peer review of "Biologic Treatment in Combination with Lifestyle Intervention in Moderate to Severe Plaque Psoriasis and Concomitant Metabolic Syndrome: Rationale and Methodology of the METABOLyx Randomized Controlled Clinical Trial"

_nutrients, 2021, doi:10.3390/nu13093015_

Round 1
Reviewer 1 Report
This is a methodology article of a trial comparing the response to secukinumab with or without additional lifestyle intervention.
Major revisions
1/ The literature cited includes many reviews. Please consider citing the original articles.
2/ Please provide the numbers of the protocol that was approved by the ethical review board
3/ Statistical analysis: will it be an intention to treat analysis? How was the randomization performed and why is the number of patient in each group different?
4/ Please provide a flow chart for the patients included
5/ Please provide the cost of the intervention
Minor revisions
1/ Extensive editing of English language and style is required, punctuation must be homogenized.
2/ The introduction is too long and should be shortened. Same for the discussion.
3/ Figure 1 is not very clear. Why would secukinumab only inhibit the IL-17 from the skin and joints and not from the fat for instance?
3/ table 2: is the increase of physical activity >30min / day an increase compare to baseline activity or is 30min in total the aim to reach? Is 15g a quantity /day?
4/Appendix: why excluding patients who received IL-17 blockers previously and TNF blockers less than a year before the trial? Why exclusing patients with HbA1c over 8%? Please define "alcohol and drug abuse",
5/ Appendix: why excluding patients starting "concomitant medication", did you have any reason to believe that these medications would differ between the intervention group and the control group?
Author Response
Reviewer 1:
This is a methodology article of a trial comparing the response to secukinumab with or without additional lifestyle intervention.
Major revisions
1/ The literature cited includes many reviews. Please consider citing the original articles.Is this possible from your side?
We have limited the citation of reviews to only the more general points which they support and have added relevant orignal articles, where possible without an extremely large increase in references being necessary, and as appropriate.
2/ Please provide the numbers of the protocol that was approved by the ethical review board
We have included the NCT03440736 number next to the EUDRAT CT number. The trial was not registered anywhere else. The internal study/protocol number is CAIN457ADE08, however we are unsure whether this would be meaningful.
3/ Statistical analysis: will it be an intention to treat analysis? How was the randomization performed and why is the number of patient in each group different?
The statistical analysis of the patient and efficacy characteristics will be performed with “all patients who were randomized into this study at baseline and received at least one dose of study treatment” (full analysis set). Drug exposure and safety analysis will be done with “all patients who received at least 1 dose of study treatment” (safety set). The full analysis set and safety set are almost identical. Both sets effectively resemble an intention-to-treat set in this trial setting.
Patients were randomized in a 1:1 ratio to one of the two treatment arms at randomization visit. Randomization was performed using sealed envelopes. The envelope assigned a randomization number to the patient, which could be used to link the patient to a treatment arm.
The difference in patient number of the two treatment arms is due to size of the randomization blocks.
Updates have been made to Figure 2 with the actual patient numbers and randomization ratio added.
4/ Please provide a flow chart for the patients included
5/ Please provide the cost of the intervention
The cost of secukinumab therapy is reimbursed by health insurance for psoriasis patients. The true representative costs of the lifestyle are difficult to establish as yet, because some aspects of the study, e.g. training of center personnel, would not need to be paid for by patients or providers in the real-world setting. Using a previous diabetes education reimbursement as example from a comparable field, the cost was approx 220 EUR/patient. However, we do not feel it is appropriate to give an estimate in the manuscript itself as prices will vary from country to country.
Minor revisions
1/ English editing/Punctuation must be homogenized.
The manuscript was drafted with the assistance of a native English-speaking Medical Writer. The revised drarft has been reviewed again for style, clarity and consistency in language and punctuation.
2/ The introduction is too long and should be shortened. Same for the discussion.
The authors feel that since this manuscript details the rationale of a novel study, and given the complexity of the interrelationship between psoriasis, inflammation and obesity, a detailed introduction is appropriate, particularly given that there is no extensive results section. We have however taken steps to edit and reduce the length of these sections where possible.
3/ Figure 1 is not very clear. Why would secukinumab only inhibit the IL-17 from the skin and joints and not from the fat for instance?
We thank the reviewer for this observation and have added the inhibition symbol where apprpopriate in Figure 1. We have also added further description of the production of IL17A by fatty tissue and its relevance to the study.
3/ table 2: is the increase of physical activity >30min / day an increase compare to baseline activity or is 30min in total the aim to reach? Is 15g a quantity /day?
The protocol does not aim to achieve an increase from baseline activity but aims for a total increase of activity of at least 30 minutes. This has been clarified in the table.
4/Appendix: why excluding patients who received IL-17 blockers previously and TNF blockers less than a year before the trial? Why exclusing patients with HbA1c over 8%? Please define "alcohol and drug abuse",
Exclusion of previous IL-17 and anti-TNF-blocker (less than a year) treated patients is done to ensure unbiased efficacy analysis. The efficacy of anti-IL-17 and anti-TNF-treatment on psoriaatic disease is well established. Beyond its impact on psoriatic skin, there is evidence for potential metabolic effects of anti-TNF and anti-IL17A-treatment as well. In this study, we aim to investigate potential metabolic effects of anti-IL-17 treatment in PsO-patients with metabolic syndrome and therefore wanted to exclude potential overhang effects of previous anti-IL-17 or anti-TNF-treatment, by excluding patients with previous anti-IL17A treatment and by requesting a 1 year wash-out period for prior anti-TNF-treatments to rule out any potential overhang effect. The length of this washout period is a standard. The reason for the different handling of prior anti-IL17 and prior anti-TNF-treatment is that totally exluding patients with prior anti-TNF-treatment would have significantly hampered recruitment of the study due to the widespread use of anti-TNF directed agents in moderate to severe Psoriais. Therefore, we have decided to allow the enrollment of anti-TNF-pretreated patients, but safely ruling out any overhang effect by a 1 year wash-out period.
In the protocol, HbA1c >8% classifies patients with uncontrolled type 2 diabetes. Uncontrolled diabetes presents a potential safety risk for patients which needs to be avoided. It is likely that the diagnosis of uncontrolled type 2 diabetes would trigger an adjustement of anti-diabetic treatment, typically by updosing of existing medication or by adding new anti-diabetic medications in order to improve glycemic control. Inclusion/Exclusion criteria need to be complied to at Visit 2. This would lead to violations of the protocol which prohibits the prescription of new/additional anti-diabetic treatments during the core study to ensure a robust analysis of the metabolic impact of the study treatment without bias by concomitant medications.
“Alcohol and drug abuse” was defined as use of alcohol/drugs causing sustainable physical, psychocolical and social harm. This would have rendered adherence to the protocol challenging and potentially introduced bias.
5/ Appendix: why excluding patients starting "concomitant medication", did you have any reason to believe that these medications would differ between the intervention groupi8. and the control group?
Patients starting cholesterol- or lipid lowering agents (e.g. HMG-CoA-inhibitors/statins, fibrates, nicotinic acid, ezetimibe, colestyramin, colestipol), antihypertensive drugs (e.g. ACE-inhibitors, angiotensin-receptor antagonists, ß-blockers, aldosterone receptor antagonists, diuretics, nitrates, calcium channel blockers, Aliskiren, Clonidin, alpha1 receptor antagonists (e.g. Doxazosin), Dihydralazin, Minoxidil, Clonidin, Moxonidin or Methyldopa) and/or glucose-lowering agents (e.g. Metformin, DPP4 inhibitors, SGLT2 inhibitors, GLP1 analogues etc.) were excluded because starting treatment with these medications during the study treatment phase would clearly impair the analysis of the metabolic impact of the study treatment. The sole effect of the lifestyle intervention together with an IL-17 A inhibition can no longer be verified in this way.
Reviewer 2 Report
The authors presented rationale and methodology for clinical trial assessing the role of lifestyle modification in patients with psoriasis and metabolic syndrome who were treated with secukinumab.
It is very interesting concept and clinically relevant.
There are some minor issues that need clarification.
- Introduction, line 48 "drosophila melanogaster" - please change to "Drosophila melanogaster"
- Introduction - while the authors present connections between inflammation and metabolism it is important to mention the concept of "metainflammation"
- Table 3, please give explanation for biomarkers abbreviations
- P1NP may be increased in patients with NAFLD which is highly prevalent in psoriasis and metabolic syndrome. Please, discuss.
- The role of IL-17 in metabolic complications of psoriasis should be presented.
Author Response
Reviewer 2: The authors presented rationale and methodology for clinical trial assessing the role of lifestyle modification in patients with psoriasis and metabolic syndrome who were treated with secukinumab. It is very interesting concept and clinically relevant. There are some minor issues that need clarification. Introduction, line 48 "drosophila melanogaster" - please change to "Drosophila melanogaster" This has been changed. Introduction - while the authors present connections between inflammation and metabolism it is important to mention the concept of "metainflammation" We have integrated the definition of meta-inflammation (p. 2) Table 3, please give explanation for biomarkers abbreviations The expansions have been added. P1NP may be increased in patients with NAFLD which is highly prevalent in psoriasis and metabolic syndrome. Please, discuss. The authors thank the reviewer for raising this interesting point, especially in regard to the interpretation of the biomarker results. The authors fully agree that P1NP is a promising future biomarker of osteoporosis risk, but its significance in psoriasis is currently not yet determined (DOI: 10.2217/17520363.2.5.495, DOI: 10.1007/s00198-019-05160-x) . There is evidence indicating that NAFLD patients have a higher risk for osteoporosis/osteoporotic fractures (DOI: 10.1097/MEG.0000000000000535, DOI: 10.1186/s12967-016-0766-3, DOI: 10.3389/fendo.2018.00408). The pathophysiological mechanisms linking NAFLD, NAFLD-risk factors (e.g. obesity and type-2-diabetes) and osteoporosis are however largely unknown. In this context, pro-inflammatory cytokines mediating chronic metainflammation (e.g. TNF-alpha, IL-17), hepatocines, adipokines (e.g. Leptin, adiponectin) and osteokines (e.g. Osteoprotegerin, Osteocalcin) are suggested as potential pathomechanistic regulators of bone metabolism (PMID: 26770315, PMID: 18251167., DOI: 10.1111/j.1365-2249.2011.04471.x, DOI: 10.1007/s00125-011-2170-0, DOI: 10.3389/fendo.2018.00408, DOI: 10.1016/j.bbrc.2005.03.210, DOI: 10.1016/j.cca.2012.11.029, DOI: 10.1002/jbmr.2083). Due to the specific characteristics of the METABOLyx study population (obesity, insulin resistance, chroic inlammatory skin disease), the biomarker analysis data will be correlated not only with clinical outcomes measures and metabolic disease markers (including fatty acids, triglycerides), but also interpreted considering standard liver function panel. Until now, however, there is little direct evidence of a potential P1NP increase in NAFLD patients with most human studies reporting decreased or unchanged levels of P1NP in NAFLD patients (DOI: 10.3389/fendo.2019.00926; DOI: 10.1016/j.diabet.2018.10.001, DOI: 10.1210/jc.2018-00176). Consistently, decreased P1NP levels were found in patients with metabolic syndrome which were positively correlated with decreased levels of sThy1/CD90 reported in these patients (DOI: 10.1097/gme.0b013e3181e39a15, DOI: 10.1126/scitranslmed.aao6806). We have added some further information on P1NP to the revised draft as requested, taking in to consideration the request to shorten the inroductory material overall. The role of IL-17 in metabolic complications of psoriasis should be presented. We fully agree that the role of IL-17 in metabolic complications of psoriasis is an interesting and important topic. The results of the METABOLyx trial will substantially add to the increasingly apparent role of IL-17. We addressed the implications of IL-17 in the manuscript as suggested. Upon completion of the analysis of the METABOLyx trial, we will have a better understanding of the effect of IL-17 inhibition in metabolic syndrome patients with psoriasis. These result will enable us to draw meaningful conclussions for the role of IL-17 and discuss in more detail. At this point, however, we believe that discussing the role of IL-17 in metabolic complications of psoriasis even further without the respective results from the presented trial would not add to the manuscript, but exceed its scope especially considering the noted length of the paper.
Round 2
Reviewer 1 Report
-